# Sericin *Ser3* Ectopic Expressed in Posterior Silk Gland Affects Hemolymph Immune Melanization Response via Reducing Melanin Synthesis in Silkworm

**DOI:** 10.3390/insects14030245

**Published:** 2023-02-28

**Authors:** Yongfeng Wang, Meijuan Shi, Jiameng Yang, Lu Ma, Xuedong Chen, Meng Xu, Ruji Peng, Guang Wang, Zhonghua Pan, Yanghu Sima, Shiqing Xu

**Affiliations:** 1School of Biology and Basic Medical Sciences, Suzhou Medical College, Soochow University, Suzhou 215123, China; 2Institute of Agricultural Biotechnology & Ecology (IABE), Soochow University, Suzhou 215123, China

**Keywords:** *Bombyx mori*, silk gland transgenic, immune melanization response, melanin synthesis

## Abstract

**Simple Summary:**

Silkworms have remarkable protein synthesis ability, which is favored by biomaterials and biomedicine researchers. The silk gland is the most concerned target tissue of sericulture. Many studies have focused on introducing exogenous functional fibrin genes into the silkworm genome, and have obtained biomedical proteins with good biological activity and multifunctional silk fibers with special properties using the silk gland bioreactor. However, the silk gland of transgenic silkworms often suffers from low vitality, stunting and other problems, and the reasons are still unknown; the biosafety of transgenic silkworms is also unknown. Therefore, we studied the effect of the recombinant *Ser3* ectopic gene expressed in the posterior silk gland on hemolymph melanization and metabolism in silkworms. The results showed that although the mutant had normal vitality in normal environments, hemolymph metabolism and immunity were significantly affected. Therefore, the results have positive significance to promote the safe assessment and development of genetically modified organisms.

**Abstract:**

The transgenesis of silkworms is an important way to innovate genetic resources and silk function. However, the silk-gland (SG) of transgenic silkworms, which is the most concerned target tissue of sericulture, often suffers from low vitality, stunting and other problems, and the reasons are still unknown. This study trans engineered recombinant *Ser3*, a middle silk gland (MSG) specific expression gene, in the posterior silk gland (PSG) of the silkworm, and studied hemolymph immune melanization response changes in mutant pure line SER (*Ser3*^+/+^). The results showed that although the mutant had normal vitality, the melanin content and phenoloxidase (PO) activity in hemolymph related to humoral immunity were significantly reduced, and caused significantly slower blood melanization and weaker sterilization ability. The mechanism investigation showed that the mRNA levels and enzymatic activities of phenylalanine hydroxylase (PAH), tyrosine hydroxylase (TH) and dopamine decarboxylase (DDC) in the melanin synthesis pathway in mutant hemolymph, as well as the transcription levels of the *PPAE*, *SP21* and serpins genes in the serine protease cascade were significantly affected. Moreover, the total antioxidant capacity, superoxide anion inhibition capacity and catalase (CAT) level related to the redox metabolic capacity of hemolymph were significantly increased, while the activities of superoxide dismutase (SOD) and glutathione reductase (GR), as well as the levels of hydrogen peroxide (H_2_O_2_) and glutathione (GSH), were significantly decreased. In conclusion, the anabolism of melanin in the hemolymph of PSG transgenic silkworm SER was inhibited, while the basic response level of oxidative stress was increased, and the hemolymph immune melanization response was decreased. The results will significantly improve the safe assessment and development of genetically modified organisms.

## 1. Introduction

The superior performance of silk fibers produced by the silkworm is widely used in textile engineering. The silkworm’s silk gland (SG) bioreactor has remarkable protein synthesis ability, which is favored by biomaterial and biomedical researchers [1,2], and can be used to produce growth factors, tumor antigen proteins, human connective tissue growth factor (CTGF), recombinant human fibrinogen (rFib) and other biomedical proteins with good biological activity [3,4,5,6]. At the same time, silk gland transgenes based on textile engineering applications have also developed multifunctional silk fibers with improved silk fiber or special properties [7,8]. 

However, there are still many unsolved problems in transgenic silkworms, including the biological security problems such as abnormal development of the silk gland, low production capacity of silk gland protein, and poor individual vitality caused by the expressions of foreign genes in the silk gland [8,9,10], or transgenic technology problems such as the location effect caused by random integration of *piggybac* transposons [11,12,13,14,15]. One example is the expression of *MaSp1* in the posterior silk gland of the silkworm, which leads to the reduction in the size of the individual and the posterior silk glands, resulting in a reduction in silk protein secretion [8]. The expression of the Pieris rapae cytotoxin Pierisin-1A (P1A) gene also leads to a shortened posterior silk gland and dysfunction, with the secretion of silk protein 79% lower than that of the wild type [9].

The problem of random transgenic integration of *piggybac* transposons has been solved using site-specific recombination technology [16,17], a GAL4/UAS dual expression system mediated by a tissue-specific promoter [18,19], and CRISPR-Cas9 gene editing technology [20,21,22]. Although some studies have recently begun to pay attention to biological security problems, such as developmental barriers and the low vitality of transgenic silkworms, there is still a lack of studies on the effects on biological security such as metabolism and immunity [10,23], despite reports that silkworm resistance was improved by the overexpression of antiviral or antimicrobial peptide-related genes specific to non-silk gland tissue when breeding silkworm varieties with high disease resistance [24,25,26].

Melanization is an important part of the immune defense system of insects including the silkworm, participating in humoral and cellular immunity [27,28,29]. The silkworm has an open structure of blood circulation, and the hemolymph is the storehouse of nutrition, energy and metabolism of silk gland cells [30]. This study used the pure line SER material of the silkworm mutant with the self-silk protein gene, which is specific for the silk gland tissue. The growth and development of silk glands and individuals did not have adverse problems, with the silk protein production capacity of the silk gland exceeding that of the wild type (WT), and the recombinant *Ser3* fragment of the mutant was not inserted into the functional gene site [1]. Through the influence mechanism of SER silkworm hemolymph melanization ability, this study discussed the influence of silk gland transgenicity on the melanization process of silkworms and the redox metabolism of hemolymph in vivo, to provide a reference at the molecular level for promoting the safe evaluation and development of transgenic silkworms.

## 2. Materials and Methods

### 2.1. Experimental Animal Preparation

The mutant pure line SER is a recombinant *Ser3* gene specifically expressed in the middle silk gland and posterior silk gland. It was created by this laboratory, and WT is its wild type [1]. The silkworm larvae were fed with fresh mulberry leaves under natural light at 25.0 °C ± 2.0 °C. At the 5L1d, 5L3d and W stages, samples of hemolymph, fat bodies and silk glands from female silkworms were collected on ice with three duplicates. Each duplicate sample used in the subsequent various determinations was from at least three larvae. Hemolymph was mixed in equal volumes, and fat bodies and silk glands were mixed in equal weights after being ground with liquid nitrogen.

### 2.2. Hemolymph Melanization Speed

The fresh 200 μL hemolymph samples were added into a sterile 96-well plate (Corining Costar, Corning, NY, USA), and the change speed of hemolymph melanization was immediately recorded at 25 °C under normal light.

### 2.3. Plasma Melanin Content

The determination and calculation of melanin content was performed as described previously [31], and was slightly modified. Hemolymph samples were transferred to sterilized centrifuge tubes (Axygen, San Francisco, CA, USA) and centrifuged at 3500 rpm at 4 °C for 5 min. Then, 100 μL of supernatant plasma was mixed with 1 mol/L NaOH containing 10% DMSO in equal volume, incubated at 80 °C for 2 h, and the OD value at A405 nm was measured with a multi-function microplate reader (Synergy HT, Biotek, Winooski, VT, USA).

### 2.4. Bactericidal Capacity of Plasma Melanism

Hemolymph samples from 5L3d larvae of wild-type silkworms were centrifuged at 3500 rpm at 4 °C for 5 min, and 400 μL of supernatant plasma was taken. A volume of 10 μL of Hank’s Balanced Salt Solution (HBSS) (Sangon Biotec, Shanghai, China) was added to the control group, and 10 μL of phenylthiourea (PT) (Adamas, Basel, Switzerland) was added to the plasma melanization inhibition group to inhibit plasma melanization; then, 100 μL of each sample was added to the 96-well plate, with each group repeated with three duplicates. The HBSS group and PT group had 20 μL of *E. coli* and *S. aureus* added, respectively. The groups were cultured at 37 °C for 90 min, and then each sample had 25 μL of 2 μg/mL propidium iodide (PI) dye solution (Sangon Biotec, Shanghai, China) added for dead bacteria staining. After staining at room temperature for 20 min in the dark, the samples were placed on a slide, and red fluorescence from the PI was observed under a BX51 fluorescence microscope (Olympus, Tokyo, Japan).

### 2.5. Real-Time Fluorescence Quantitative Polymerase Chain Reaction (PCR)

The total hemolymph RNA in hemocytes and fat bodies were extracted with an RNAiso Plus Kit (Takara, Dalian, China) and reverse transcribed into cDNA. The *Rp49* gene was used as internal control. Then, 20 µL of reaction solution was prepared according to the instructions of the Takara kit, and the reaction procedure was set at 95 °C for 30 s, 95 °C for 30 s, 60 °C for 30 s and 95 °C for 5 s; a total of 40 cycles were performed using an ABI StepOnePlus™ real-time PCR system (Ambion, Foster City, CA, USA). The primers used in the study are listed in Appendix A.

### 2.6. Enzyme Activity

Phenoloxidase (PO) activity was measured by mixing 10 µL of plasma with 180 µL of PBS preheated to 37 °C, and 10 µL of 0.01 mol/L L-Dopa was added into the 96-well plate and mixed; the OD value was measured at 490 nm at 37 °C with a multi-function microplate reader, and then measured every 2 min for a total of 10 times. The calculation process was performed as described previously [27].

The activities of PAH, TH and DDC were measured via ELISA, according to the kit instructions (Kete Biotechnology, Nanjing, China). The standard was diluted gradiently, and the plasma sample was diluted five times, with 50 µL of each added to the enzyme-labeled coated plate, and a 50 µL sample dilution was added to the blank well for blank control. After incubation at 37 °C for 30 min, the tray was washed five times. A volume of 50 µL of enzyme-labeled reagent was added to all wells except the blanks and incubated at 37 °C for 30 min, then washed again. Volumes of 50 µL of each of chromogenic reagents A and B were added to each well successively and mixed. The reaction was kept away from light at 37 °C for 10 min, after which 50 µL of termination solution was added, and the OD value was measured at 450 nm with a multi-functional microplate reader.

The activities of SOD, CAT and GR were measured using the SOD assay kit, CAT assay kit and the GR assay kit, according to the kit instructions (Jiancheng Bioengineering Institute, Nanjing, China). The plasma of silkworms was collected according to the method described above, and then the corresponding reagents were added to incubate for a specified time. The multifunctional enzyme marker was used to measure the OD value of SOD at 450 nm, CAT at 405 nm and GR at 340 nm.

### 2.7. Hemolymph Protease Inhibitor Activity

The hemolymph of silkworms was collected on ice and centrifuged at 3500 rpm at 4 °C for five min to remove plasma; then, 150 µL of lysate and 1.5 µL of phenylmethylsulphonyl fluoride (PMSF) (1 mmol/L) were added to the bottom hemocyte precipitate. The hemocytes were then spun at 1500 rpm at 4 °C for 4 h to fully lysate and extract proteins. 

Each sample was subjected to SDS-PAGE gel electrophoresis with 100 µg of total protein. After electrophoresis, the gel was cleaned with ultrapure water for 1 min. The protease solution was then added and incubated at 37 °C and 50 rpm in the dark for 20 min. The protease solution was discarded, and the gels were cleaned twice. The dye solution and matrix solution were then added and incubated at 37 °C in the dark for 30 min. Finally, the dye solution was discarded, the gel was cleaned, and an image was taken with a camera (Canon, Tokyo, Japan).

### 2.8. Plasma Redox Substrate Content

The levels of total antioxidant capacity (T-AOC), inhibition ability of anti-superoxide anion free radicals (ASAFR), reduced glutathione (GSH) and H_2_O_2_ in the plasma were measured with the total antioxidant capacity assay kit, the inhibition and product superoxide anion assay kit, the reduced glutathione (GSH) assay kit and the hydrogen peroxide assay kit respectively, according to the instructions (Jiancheng Bioengineering Institute, Nanjing, China). The plasma of silkworms was collected according to the method described above, and incubated with corresponding reagents for 5–10 min. The multi-functional enzyme marker was used to measure the OD value of T-AOC, GSH and H_2_O_2_ at 405 nm, and the OD values of the content of the inhibition and production of superoxide anion radical at 550 nm.

### 2.9. High Temperature Resistance

Male larvae of 4L2d and 5L4d were placed in an incubator (Jiangnan, Ningbo, China) with a high temperature of 40 °C and a relative humidity of 84% for 3 h without touching mulberry leaves. Then, the was transferred to an incubator at 25.0 °C ± 2.0 °C under normal natural light conditions, and fresh mulberry leaves were fed. The number of surviving silkworms was counted every 24 h, and the development status was observed. Each group had 3 replicates, and each replicate had 20 silkworms.

### 2.10. Data Analysis

All statistical analyses were performed using GraphPad Prism 8 software (v8.0.2, GraphPad, San Diego, CA, USA). The values of the data of WT at the age of 5L1d were normalized to about 1. Data were presented as mean ± SD; the Holm–Sidak multiple *t*-test and two-way ANOVA multiple comparisons were used for the comparison between the two groups, and the *p* value obtained was the adjusted *p* value. *n* = 3 samples, and each sample came from three different silkworm individuals.

## 3. Results

### 3.1. The Recombinant Ser3 Gene Expressed in the Posterior Silk Gland Affects the Hemolymph Immune Melanization Response of Silkworms

The fifth instar (5L) of silkworm larvae is a period of mass expression of the silk protein gene, rapid synthesis of silk protein, and secretion into the silk gland cavity. In order to study the effect of the recombinant *Ser3* gene expression in the posterior silk gland (PSG) on the immune resistance of silkworms, the melanization of silkworm hemolymph, which is closely related to humoral and cellular immunity, was investigated. From the early stage (5L1d) to the late stage (5L5d) of the 5L of the larva, the hemolymph of the WT silkworm began melanization after being exposed to the air for about 20 min. The starting time of melanization had little correlation with the age of the 5L (Figure 1a). The hemolymph of the SER silkworm took more than 2.5 h in vitro to approach the melanization start of the WT group, and the melanization rate was about five times longer. 

By comparing the difference in the rate of hemolymph melanization among different silkworm strains, melanization was a common phenomenon. Although there were differences in the time of hemolymph melanization in vitro of 11 representative silkworm varieties or germplasm resources in 5L3d larvae, they all initiated melanization within 5–10 min; only one PSG transgenic silkworm strain TBH started melanization after 30 min (Appendix A), proving that the transgenic expression of exogenous proteins in PSG could significantly reduce the rate of melanization in the hemolymph of larvae.

The changes in melanin content in the hemolymph of transgenic silkworms were further investigated. The results showed that the melanin content of mutant SER was significantly lower than that of WT hemolymph in 5L1d and 5L3d, but there was no statistically significant difference between SER and WT at the wandering stage (W) (Figure 1b); meanwhile, the plasma melanization speed was still significantly slower than that of WT (Figure 1a). It is worth noting that the melanin content in hemolymph of the fifth instar larvae of SER was reduced, which was consistent with the very high expression period of the silk gland *Ser3* gene (PSG + MSG) superposed by PSG cells caused by transgenic change (Figure 1c). In the W stage, the transcription level of the *Ser3* gene in the SER silk gland was reduced to a level lower than that of WT (Figure 1c), and the melanin content in hemolymph also recovered to that of the WT level. The results indicated that the expression of the recombinant *Ser3* gene in the PSG of the mutant in the early and middle stages of the 5L affected the melanin production in hemolymph.

To identify the germicidal efficacy of melanization of WT silkworm hemolymph, *E. coli* and *S. aureus* were co-cultured with normal plasma (HBSS) and non-melanized plasma with added phenylthiourea (PT), as shown in Figure 1d. The HBSS group was accompanied by rapid plasma melanization, and propidium iodide (PI) staining showed that bacteria were dead in large numbers and highly clustered. However, the plasma in the PT group did not darken for a long time, and there were few dead bacteria and no serious bacterial aggregation. The results showed that melanization played an important role in the immune response of silkworm after the invasion of a pathogen.

To further study the effect of recombinant *Ser3* expression in PSG on the melanin synthesis pathway, the study focused on four key enzymes, including phenylalanine hydroxylase (PAH), tyrosine hydroxylase (TH), dopa decarboxylase (DDC) and phenol oxidase (PO). PAH converts phenylalanine into tyrosine, TH hydroxylates tyrosine into dopamine hydroxylase (TH), DDC converts dopamine into dopamine decarboxylase and PO converts dopamine into dopamine pigment. An enzyme activity survey showed that the activities of PAH, TH and DDC in the hemolymph of SER were higher in 5L1d and 5L3d than in the WT group. However, at the W stage, although PAH was still higher than WT, DDC was lower than WT (Figure 2a–c). It is noteworthy that the PO activity of the mutant SER remained at a very low level throughout the period, maintaining around 10% of WT (Figure 2d); meanwhile, the activities of the other three enzymes in the mutant showed an increasing trend compared with those in the WT group, but were less than one times higher than those in the WT group (Figure 2a–c). Gene transcription levels also showed that that the mRNA level changes of four enzyme genes in the 5L larvae were consistent with their enzyme activity levels (Figure 2e–h). The expressions of the *PAH*, *TH* and *DDC* genes in hemocytes of the SER were significantly up-regulated compared with those in the WT group, except at the W stage (Figure 2e–g), while the *PPO1* and *PPO2* genes were significantly down-regulated (Figure 2h and Appendix A), showing a time-inhibitory effect. However, the expression level of the transcription factor *Lozenge* gene in the hemolymph of SER silkworm was significantly up-regulated (Appendix A). It showed that the rate-limiting step of SER melanin synthesis was the key enzyme PO of the final reaction, and its gene transcription level was down-regulated with the enzyme activity significantly inhibited; this resulted in a decrease in melanin content in hemolymph and its slowed melanization.

### 3.2. Mutant Silkworm Affects Hemolymph Melanization by Changing the Serine Protease Cascade

In order to study the reason for the decrease in PO enzyme activity in the rate-limiting step of melanin synthesis in mutant hemolymph, the gene transcription levels of serine protease 21 (SP21) and serine protease phenoloxidase-activating enzyme (PPAE) that finally activate PPO were investigated. The results showed that the mRNA levels of SP21 (Figure 3b) and PPAE (Figure 3c) in SER hemolymph were up-regulated compared with those in WT silkworm at 5L1d and 5L3d, but down-regulated at the W stage; furthermore, the up-regulated and down-regulated changes were less than one-fold. It is worth noting that the mRNA levels of these two genes (Appendix A) were significantly up-regulated compared with those of WT in the main synthetic tissue fat body during the whole 5L period. The *PPAE* gene was up-regulated more than 200 times in 5L3d, and also significantly up-regulated in the W stage.

Further investigation of the changes in the transcription levels of serine protease inhibitor family genes showed that the *Spn5*, *Spn6* and *Spn32* genes were significantly up-regulated in SER hemolymph (Figure 3) and fat bodies (Appendix A) in 5L1d and 5L3d, similar to *PPAE* and *SP21*. The extent of up-regulation was greater in fat bodies, which were four-, 550- and 100-times higher than the WT group in 5L3d, respectively, and the *Spn32* gene was still up-regulated nearly eight times compared with the WT group in the W stage. The transcription levels of the *Spn6* and *Spn32* genes in SER fat bodies were sharply up-regulated by more than 100 times during the critical period of *Ser3* gene transition from high to low expression in the silk gland of 5L3d mutant (Appendix A), but the *PPAE* gene that ultimately activated PPO was also up-regulated by 200 times (Appendix A), showing a two-way extreme abnormal consumptive regulation of proenzyme PPO activation and negative regulation of a key rate-limiting enzyme in melanin synthesis. 

Staining analysis of the inhibitory activity of serine protease inhibitors in hemolymph showed that both WT and SER hemolymph had stable inhibitory activity bands for α-chymotrypsin, subtilisin A and proteinase K (Figure 3g–i), but no inhibitory activity bands for trypsin (Figure 3j). Compared with the difference in activity represented by the area of stained bands, SER hemolymph showed stronger inhibitory activity against protease K than WT, especially in 5L1d (Figure 3i). It was suggested that the expression levels of serine protease inhibitor genes *Spn5*, *Spn6* and *Spn32* were significantly up-regulated in the mutant, which may have been mainly through the enhanced inhibitory activity of their products on proteinase K, which further affected the expression of PPO cascade genes. 

### 3.3. Mutant Silkworm Affects Hemolymph Melanization by Changing the Level of Hemolymph Redox Metabolism

In order to further explore the relationship between the changes in hemolymph melanization ability and redox metabolism of SER silkworms, the redox ability of hemolymph and the expression levels of its related genes were investigated. The results showed that the hemolymph total antioxidant capacity (T-AOC) of SER silkworms during the whole 5L larval stage was significantly increased (Figure 4a), and inhibited the ability of anti-superoxide anion free radicals (ASAFR) that were also 30 to 40% higher than those of WT silkworms over the same period (Figure 4b). The content of H_2_O_2_ in hemolymph of mutant silkworm (Figure 4d) was always significantly lower than that of WT silkworm, but it kept rising in the 5th instar, and increased by nearly 50% in the W stage compared with 5L1d. It is worth noting that the enzyme activity level of SOD involved in oxygen free radical scavenging in the hemolymph of SER silkworms was lower than that of WT (Figure 4c); moreover, the content of glutathione (GSH), which both scavenges oxygen free radicals and H_2_O_2_, was significantly lower than that of WT silkworm in the same period (Figure 4f). The activity of glutathione reductase (GR), which generates GSH, had only a trace amount (Figure 4g), but the activity of the CAT enzyme was significantly higher than that of WT silkworms over the same period (Figure 4e).

The investigation results of gene mRNA levels showed that although the *TPX* (Figure 4h) and *CAT* (Figure 4i) genes that cleared H_2_O_2_ in SER showed no significant differences from those in WT silkworms in the early and middle stages of the 5L, they showed opposite changes at the W stage; the down-regulated expression amplitude of the *TPX* gene was close to the up-regulated expression amplitude of the *CAT* gene. It is worth noting that the transcription level of *Mn-SOD* of the SER silkworm was significantly up-regulated compared with that of WT in the middle and late stages of the 5L larvae (Figure 4j), while *Cu*/*Zn-SOD* was always maintained at a similar level to that of WT (Figure 4k), which was inconsistent with the enzyme activity level. It is hypothesized that there may be inhibitory molecules affecting the activity of SOD in hemolymph. These results indicated that the ectopic expression of the recombinant *Ser3* gene in PSG led to an increased oxidative stress response level in hemolymph, and increased the antioxidant capacity in body fluids.

### 3.4. Mutant Silkworm Affects the Immune Resistance of Silkworms

In order to verify whether humoral immunity was affected in transgenic mutants, we further investigated transcription levels of antimicrobial peptides (AMPs) genes in fat bodies. The results showed that the mRNA levels of moricin, cecropin A, lebocin and attacin were significantly up-regulated in 5L1d and 5L3d (Figure 5a–d). At the W stage, moricin, lebocin and attacin were down-regulated, except cecropin A, which was still up-regulated, presumably due to autoimmune regulation. These results showed that the ectopic expression of the *Ser3* gene in the PSG induced the expression of AMPs.

In our previous study, we found that there was no significant difference between the growth and development of SER silkworms and WT under normal conditions [1]. In order to explore whether there were differences under stress conditions, we investigated the growth and development of 4L2d and 5L4d silkworm larvae after 40 °C high temperature shock for 3 h (Figure 5e–h). The results showed that the survival rate of SER larvae significantly decreased after high temperature shock at 4L2d, and the number of SER individuals that developed into adults decreased, but there was no significant difference (Figure 5e). After high temperature shock at 5L4d, the survival rate of SER larvae had no significant effect, but all died in the pupal stage, while 11.67% of WT pupae successfully developed into adults (Figure 5f). The survival curve showed that both WT and SER silkworms began to die at 96 h after high temperature shock at 4L2d; the number of dead individuals increased rapidly from 120 h, and the mortality rate of the SER group was higher than that of the WT group. Finally, 26.67% and 8.33% silkworms in the WT and SER groups spun cocoons, respectively (Figure 5g). At the same time, it was found that after the high temperature shock at 4L2d, the proportion of SER silkworms from molted and molting larvae, and the proportion of non-molting and dead larvae increased compared with WT (Figure 5h). The results showed that high temperature stress significantly affected the growth and development of SER silkworms, suggesting that the basic resistance of SER silkworms was different from that of WT silkworms.

## 4. Discussion

### 4.1. The Recombinant Ser3 Expression in Posterior Silk Glands Affects Hemolymph Metabolism

Insect melanization, including that in the silkworm, is activated by the prophenoloxidase (PPO) cascade, and mediated and regulated by serine proteinase (SP) and serine protease inhibitors (serpins) [32,33,34]. Phenoloxidase (PO) is a key rate-limiting enzyme [35], which is mainly secreted by hemocytes called oenocytoids in the hemolymph of the silkworm [36]; it exists in the form of PPO1/PPO2, is activated by SP [37,38] and is negatively regulated by serpins such as *Spn5*, *Spn6* and *Spn32* [32,39,40]. The expression of serpin genes in fat bodies was higher than that in other tissues [40]; our study also shows consistent results. In addition to serpins, PO activity is also regulated by clustering proteins and transcription factors [41,42,43,44]. After an overexpression of transcription factor *Lozenge* in silkworm, the expressions of *PPAE*, *PPO1* and *PPO2* genes were up-regulated, and the melanization rate of hemolymph was accelerated when exposed to air, while the melanization rate of hemolymph was significantly slowed down after interference with *Lozenge* [42]. In this study, the transcription levels of *PPO1* and *PPO2* were significantly down-regulated, but the expression level of the *Lozenge* gene was significantly up-regulated. These contradictory changes in a series of key enzyme genes and enzyme activities of melanin synthesis, accompanied by the simultaneous high expression of the exogenous gene *Ser3* of the silk gland transgene, showed that the production of melanin in the silk gland transgene mutant was affected by many complex aspects, and ultimately led to the reduction in melanin level in hemolymph. 

The production of melanin and the rate of hemolymph melanization in silkworm were also closely related to redox metabolism in vivo. Previous studies have shown that eliminating ROS and improving the antioxidant level of cells can effectively inhibit UV-induced melanin formation [45,46,47], and the excessive production of ROS under oxidative stress leads to the massive production and abnormal metabolism of melanin [48,49]. The results of this study showed that the antioxidant capacity of hemolymph in the SER silkworm was enhanced, and the content of H_2_O_2_, which is one of the ROS, was also significantly decreased; this suggests that the decrease in melanin content may have been related to ROS levels. In addition, the results of mRNA levels and enzyme activity for CAT and SOD were not consistent. We speculate that the inconsistency may have been due to the fluctuation of weak difference values on the one hand, while on the other hand, the oxygen free radical scavenging function of SOD had other means of compensating.

### 4.2. The Recombinant Ser3 Expression in the Posterior Silk Gland Affects the Immune Ability of Silkworms

Studies have shown that insect melanization can isolate and wrap pathogens, and promote wound healing [50]. The melanization reaction is thought to strengthen nodules, provide ROS to kill microbes and to conduct immune defense together with antimicrobial peptides (AMPs) [51]. The PPO cascade regulating melanin production is a unique immune defense response of arthropods, including silkworm. The immune protein PO is involved in the immune defense response of the body, and the serpin gene family is also involved in regulating the immune response [27,28,29,33,39,40,52], where Spn5 can inhibit its activity by combining with serine proteases HP6 and HP21, and down-regulate AMPs expression and PPO activation [32]. However, AMPs expression is not only regulated by serpins, but also by Imd and Toll signaling pathways [27]. The down-regulation of AMPs gene expression levels downstream of the Imd pathway in *Drosophila* can significantly enhance heat tolerance and prolong life span [53]. When the ROS levels of silkworms decreased, it also improved their high temperature resistance [54].

In our previous study, it was found that the survival ability of SER larvae infected with *E. coli* and *S. aureus* was higher than that of WT [1], which may be related to the significantly increased AMPs level of SER silkworms. In addition to the survival and development of insects being closely related to bacteria and other microorganisms, their physiology and metabolism are also very sensitive to temperature. High temperatures can cause mitochondrial disorder, cell protein denaturation, and eventually lead to cell dysfunction and death [55,56]. Studies have shown that insect resistance to high temperature is regulated by redox signaling pathways [57,58]. A moderate dose of H_2_O_2_ injected into *Mythimna separata* larvae consistently increases ROS and antioxidant activity in hemolymph, leading to a significant increase in survival after lethal heat stress [58]. In this study, the redox metabolism of the mutant was significantly affected, the total antioxidant capacity was increased, but the contents of H_2_O_2_ and GSH were significantly reduced. Therefore, we speculated that the reduction in high temperature resistance was closely related to disorder in redox metabolism.

It can be seen that the expression of the recombinant *Ser3* gene specifically expressed by autologous MSG in PSG transgenic silkworms can avoid the poor growth and development of silk gland tissues and individuals that may occur in PSG transgenic silkworms [1]; however, the basic stress response still appeared, which led to the decline in melanin synthesis and melanization immune ability in hemolymph (Figure 6). We speculate that there may be the following reasons for this phenomenon. Firstly, SER3 protein is an exogenous water-soluble protein that is different from the original fibrin such as Fib-H for PSG, and directly affects the silk protein interaction and secretory transport of silk gland cells and tissues. On the other hand, the recombinant Ser3 gene is highly activated by the Fib-H promoter in PSG, which changes the original supply ratio of amino acids of silk fibroin protein synthesis raw materials in PSG cells [1], even if the change is weak. The nutrients and energy of silk gland cells come from hemolymph, which is also the source of raw materials for silk protein synthesis. Therefore, the ectopic high expression of SER3 in PSG affects the hemolymph immune melanization response, which does not refer to the effect of Ser3 fusion protein products expressed in the posterior silk gland itself, nor is the effect on immunity simply reduced or increased, but there is a change in the immune balance state. These results suggest that it is necessary to study the in-depth biological safety of transgenic exogenous protein mutants for the production of target silk glands and other highly specialized tissues.

## 5. Conclusions

The recombinant sericin SER3 specific to the middle silk gland was significantly expressed in the posterior silk gland of SER silkworm larvae, which also significantly affected the expressions of antimicrobial peptides and melanization. Meanwhile, it affected the temporal and spatial expression profile of the silkworm genome, resulting in changes in transcription factor regulation, cleavage of proenzyme activity, and oxidative metabolism during melanin synthesis, further leading to significant changes in hemolymph metabolism, and ultimately reducing the melanin synthesis capacity of hemolymph.

## Figures and Tables

**Figure 1 insects-14-00245-f001:**
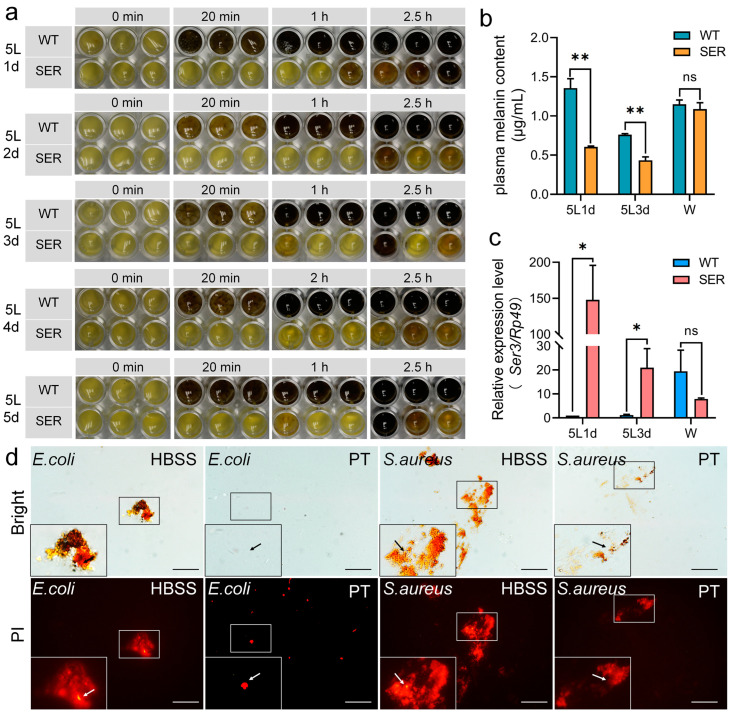
Reduced hemolymph immune melanization response of transgenic silkworm SER. (**a**) Melanization rate in vitro and (**b**) melanin content in hemolymph of 5th instar larvae. (**c**) Transcription levels of the *Ser3* gene in silk glands. *Rp49* was the internal reference gene. SER, a transgenic mutant. WT, wild type. 5L1d and 5L3d represent the 5th instar 1d and 3d larvae, respectively. W, wandering stage. * *p* < 0.05; ** *p* < 0.01; ns, no significant difference between the two groups. *n* = 3 biological repeats. (**d**) PI staining was used to investigate the germicidal efficacy of plasma melanization of 5L3d larvae. PI, propidium iodide, dyes the dead bacterial nucleus red. HBSS, Hank’s balanced salt solution. PT, phenylthiourea. Bar = 200 μm.

**Figure 2 insects-14-00245-f002:**
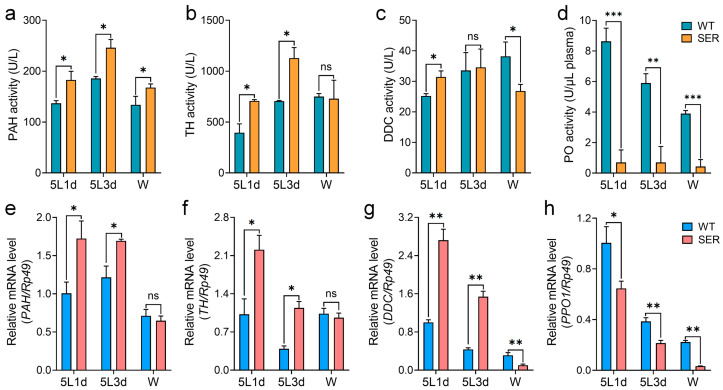
Changes in key enzyme activities and gene levels for melanin synthesis in hemolymph of transgenic silkworms. (**a**–**d**) Enzyme activity levels. (**e**–**h**) Gene transcription levels. PAH, phenylalanine hydroxylase. TH, tyrosine hydroxylase. DDC, dopa decarboxylase. PO, phenoloxidase. PPO1, prophenoloxidase 1. *Rp49* was the internal reference gene. * *p* < 0.05; ** *p* < 0.01; *** *p* < 0.001; ns, no significant difference between the two groups. *n* = 3 biological repeats.

**Figure 3 insects-14-00245-f003:**
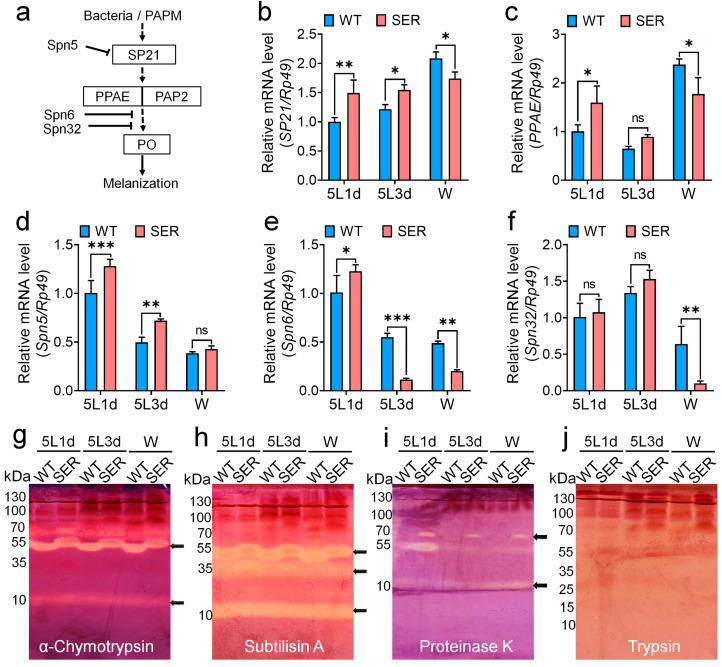
Changes in transcription levels of genes related to the serine protease cascade pathway in hemolymph of transgenic silkworms. (**a**) Schematic diagram of melanization regulation pathway. (**b**–**f**) qPCR was used to investigate the relative transcription levels of genes. *PPAE* and *SP21* are the positive regulatory genes that eventually activate PPO into PO; *Spn5*, *Spn6* and *Spn32* are the main member genes of the serine protease inhibitor family that are reversely regulated; and *Rp49* was the internal reference gene. (**g**–**j**) Activities of serine protease inhibitors in hemolymph. (**g**) α-chymotrypsin. (**h**) Subtilisin A. (**i**) Proteinase K. (**j**) Trypsin. * *p* < 0.05; ** *p* < 0.01; *** *p* < 0.001; ns, no significant difference between the two groups. *n* = 3 biological repeats.

**Figure 4 insects-14-00245-f004:**
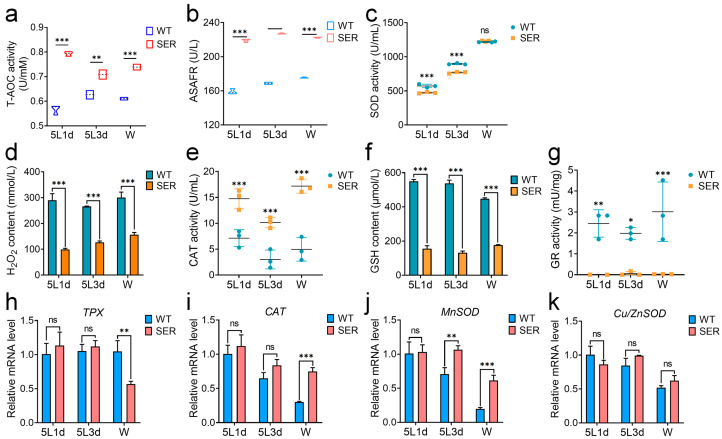
Changes in antioxidant capacity of SER hemolymph of transgenic silkworms. (**a**) T-AOC, total antioxidant capacity. (**b**) ASAFR, inhibition ability of anti-superoxide anion free radicals. (**c**–**g**) Changes in substance content and enzyme activities related to oxidative metabolism in hemolymph. (**c**) SOD activity. (**d**) H_2_O_2_ content. (**e**) CAT activity. (**f**) GSH content. (**g**) Glutathione reductase (GR) activity. (**h**–**k**) Changes in transcription levels of redox metabolism-related genes in hemolymph. (**h**) TPX. (**i**) CAT. (**j**) Mn-SOD. (**k**) Cu/Zn SOD. Rp49 was the internal reference gene. * *p* < 0.05; ** *p* < 0.01; *** *p* < 0.001; ns, no significant difference between the two groups. *n* = 3 biological repeats.

**Figure 5 insects-14-00245-f005:**
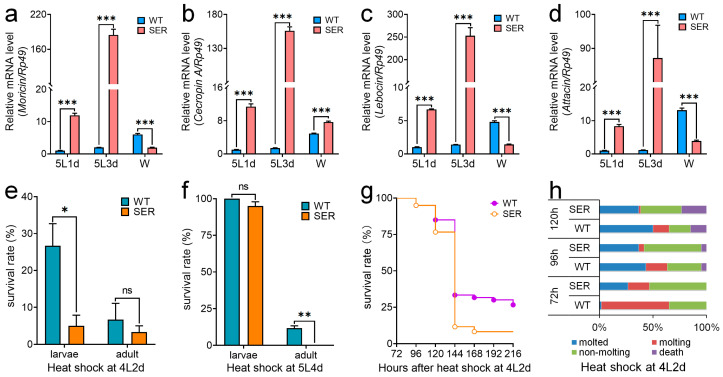
Recombinant *Ser3* expression in the posterior silk gland affected the immune resistance of silkworms. (**a**–**d**) The transcription level of antimicrobial peptide gene in fat body. (**a**) *Moricin*. (**b**) *Cecropin A*. (**c**) *Lebocin*. (**d**) *Attacin*. *Rp49* was the internal reference gene. (**e**,**f**) Survival rates of silkworms in 4L2d and 5L4d after high temperature shock. (**g**) Survival curve of silkworms in 4L2d after high temperature shock. (**h**) The growth and development of 4L2d silkworms after high temperature shock. * *p* < 0.05; ** *p* < 0.01; *** *p* < 0.001; ns, no significant difference between the two groups. *n* = 3 biological repeats.

**Figure 6 insects-14-00245-f006:**
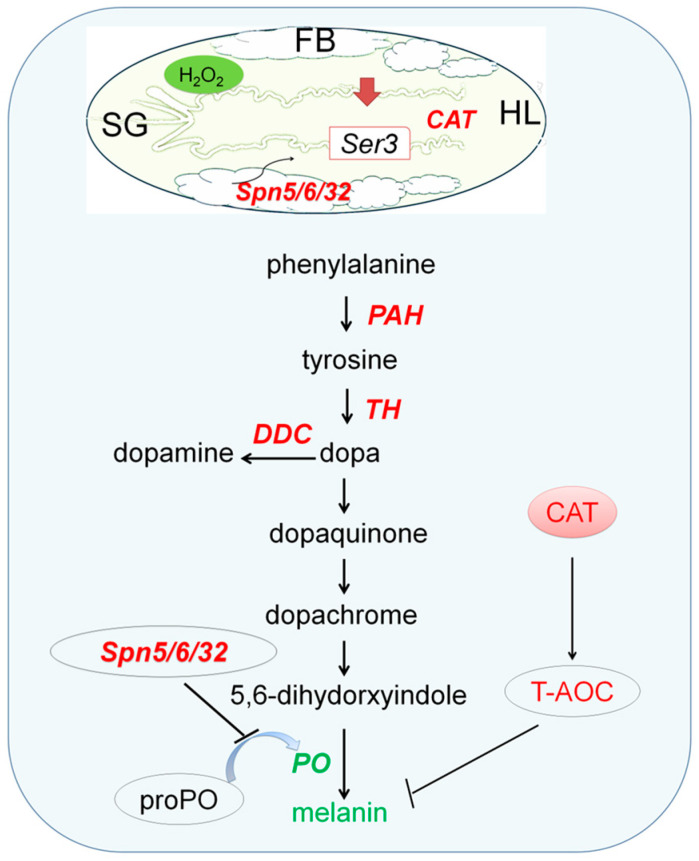
Impacts of the recombinant *Ser3* expression in posterior silk glands on the hemolymph immune melanization response of silkworms. SG, silk gland. FB, fat body. HL, hemolymph. Red means up-regulation and green means down-regulation.

## Data Availability

All of the available data are produced or analyzed with this manuscript.

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
