# Peer review of "Sericin Ser3 Ectopic Expressed in Posterior Silk Gland Affects Hemolymph Immune Melanization Response via Reducing Melanin Synthesis in Silkworm"

_insects, 2023, doi:10.3390/insects14030245_

Round 1

Reviewer 1 Report

In this paper, the hemolymph immune and melanization of transgenic materials with ectopic expression of ser3 gene in the posterior silk gland of silkworm were tested and the related genes were analyzed. It was found that the hemolymph melanization ability of individuals decreased with ectopic expression of ser3, and the melanin content decreased accordingly; Further, the gene expression and enzyme activity of its melanin synthesis pathway were detected, and it was found that its key PO gene, both protein activity and gene expression, were significantly inhibited; The author further explored the reason of its inhibition and found that the serpin gene of its pathway was significantly up-regulated. The author also detected the antioxidant active substances in hemolymph and found that there were obvious changes; It was also found that the expression of antibacterial peptide gene in fat body was also significantly up-regulated. The phenomenon revealed in this study is very interesting and surprising. It is difficult to imagine that the ectopic expression of a large protein that is not expressed in the posterior silk gland will cause such a large change in the hemolymph of the silkworm. This phenomenon is indeed worth the attention of researchers who study transgenic. Before this article is accepted, there are still several issues that need to be further confirmed,improved and explained by the author.

1.      The author mentioned that Ser3 protein was expressed by the promoter specific to the posterior silk gland. For whether the transferred ser3 gene was determined to be specifically expressed in the posterior silk gland? The author should conduct a different tissue expression test, including fluorescent quantification at mRNA level and WB at protein level with antibody of Ser3. If it has been tested, please provide corresponding documents. These results are crucial to the full text! In fact, the so-called silk gland specific genes, such as silk fibroin gene, will have a small amount of expression in other tissues. Will the transferred ser3 protein be expressed in other tissue such as fat body and secreted into the hemolymph with a small amount? This protein is a large protein, which is recognized as a foreign substance, thus triggering the immune response and oxidative stress response of the hemolymph.

2.      The author should analyze the insertion site of transgenic gene of Ser3 . In the melanization pathway, the expression and activity of PAH, TH and DDC were significantly increased in the early stage of the fifth instar, but the activity of PO gene and protein was significantly decreased, which was very interesting. Although the author investigated serpin5,6, it was found that its up regulation might inhibit PO activity, thus affecting its expression. However, the author should also rule out that the insertion of transgene into the vicinity of the gene will destroy its transcriptional activity, resulting in the reduction of its expression and activity. Moreover, both PPO1 and PPO2 are on chromosome 16. If they are inserted close to them, they may affect the transcription and protein activity of these two genes.

3.      In Figure 1a, the author provides the results of hemolymph melanization investigation in the 5L1d. In Figure 1b and Figure 1c, the author selects the quantitative results of melanin content and Ser3 expression in the posterior silk gland in the three periods of 5L1d, 5L3d and W. The author should present the results of hemolymph melanization investigation in the three periods in Figure 1a, because the melanin content and gene expression in the period of W did not change significantly. For the first two periods, it is a very important result as a good period comparison. At the same time, please supplement the unit of melanin content determination in the vertical coordinate of figure 1b.

4.      Since the author transferred Ser3 gene, it is suggested to mark Ser3 in the ectopic expression individuals in the figures and the full text, which is more accurate. The current sericin proteins, including Ser1, Ser2 and Ser3 and the recently discovered Ser4 and Ser5.  Ser are easy to be misunderstood for Ser is refer to sericin.

5.      Most of the fluorescence quantitative results in this paper are about 1 in WT at the age of 5L1d. Did the author normalize the data of WT at the age of 5L1d in addition to RP49? If yes, please supplement the calculation method in the method. At the same time, in the material method section, can there be a reference for the determination method of melanin? If yes, please list. If not, please provide the standard curve established by the melanin standard as a supplemental figure.

6.      In terms of writing, the discussion part of this article repeats a lot of the results, and it is suggested to make corresponding deletion. At the same time, a very important result, the expression of AMP and the statistical results of individual survival rate, are also discussed as Fig.S4, which is recommended to be placed in the results section and explain the paradoxical change in humoral immune resistance in discussion.

7.      In the discussion, Please the author to speculate why the ectopic expression of Ser3 in the posterior silk gland will cause a strong immune response and oxidative stress reaction in the hemolymph, and how is this signal transmitted to the hemolymph? Another curious question is why the author wants to study the hemolymph melanization investigation at the begin?

Author Response

Thank you very much for your comments concerning our manuscript. Those comments are all valuable and very helpful for revising and improving our manuscript, as well as the important guiding significance to our researches. We have tried our best to improve the manuscript and have made a lot of changes which we hope meet with approval. 

Reviewer 2 Report

The authors investigated the mechanism of abnormal melanization in SER silkworms and found that the activity and gene expression of key enzymes related to melanization were altered in the hemolymph. Additionally, they also discovered that the capacity for anti-oxidative stress differed between normal and SER silkworms. The study is well-designed and well-structured. However, several data and revisions are required for the acceptation.

1. Fig. 4; The results of mRNA levels and enzyme activity for CAT and SOD are not consistent. The authors should discuss this paradoxical.

2. Lines 366-367; The authors described that "the antioxidant capacity of hemolymph in the SER silkworm was enhanced, and the ROS level such as H2O2 content was also significantly decreased". However, SOD activity was decreased in the hemolymph of SER silkworm, suggesting that superoxide is increased in the sample. The authors should not discuss ROS as a whole, but should discuss each individual ROS. Therefore, the level of superoxide should also be measured.

3. How did the authors perform qRT-PCR? SYBR Green or TaqMan? The authors should describe the method.

4. Lines 173-178; The authors should indicate the error bars as SD.

5. Fig. 3g-j; The authors should add the molecular weights as a marker.

6. Line 163; add a spae. "ofanti-"

Author Response

Thank you very much for reviewing our manuscript and giving such a positive opinion. Your comments are all valuable and very helpful for revising and improving our manuscript, as well as the important guiding significance to our researches. We have tried our best to improve the manuscript and have made a lot of changes which we hope meet with approval.

Round 2

Reviewer 1 Report

The author has well answered all questions. At the same time, the author has supplemented the data where the article needs to be verified and improved, and made appropriate adjustments in the discussion section. However, I mentioned last time that it was need a WB verification in different tissues, especially in fat bodies, with Ser3 antibodies. It is ruled out that Ser3-OE line will express a small amount of Ser3 protein in fat bodies, which will cause immune reaction and oxidative stress reaction. Although the author has proved that the promoter used is only expressed in the posterior silk gland with several reference. However, it is not as convincing as the author to make a verification with his own transgenic materials. Genetically modified is a controversial topic, especially genetically modified products. If it is true that, as the author said, the specific expression of a foreign protein in an organ will seriously affect the immune ability of the organism, which will become another controversial topic, making the development and application of transgenic animals, plants and products even more difficult. Therefore, the author is requested to treat this issue carefully.

Author Response

We sincerely thank this reviewer for his/her careful, responsible and scientific evaluation, which is very conducive to the improvement of the manuscript and our understanding of relevant research. Please see the attachment for the point-by-point response to the reviewer’s comments.

Reviewer 2 Report

The manuscript is acceptable for the publication in its current form.

Author Response

We would like to express our great appreciation to you for your affirmation. Thank you again for your comments on our manuscript. These comments are valuable, helpful for revising and improving our manuscript, and also have important guiding significance for our research.